# UNDERSTANDING GROUNDED LANGUAGE LEARNING AGENTS

## ABSTRACT

Neural network-based systems can now learn to locate the referents of words and phrases in images, answer questions about visual scenes, and even execute symbolic instructions as first-person actors in partially-observable worlds. To achieve this so-called *grounded language learning*, models must overcome certain well-studied learning challenges that are also fundamental to infants learning their first words. While it is notable that models with no meaningful prior knowledge overcome these learning obstacles, AI researchers and practitioners currently lack a clear understanding of exactly *how* they do so. Here we address this question as a way of achieving a clearer general understanding of grounded language learning, both to inform future research and to improve confidence in model predictions. For maximum control and generality, we focus on a simple neural network-based language learning agent trained via policy-gradient methods to interpret synthetic linguistic instructions in a simulated 3D world. We apply experimental paradigms from developmental psychology to this agent, exploring the conditions under which established human biases and learning effects emerge. We further propose a novel way to visualise and analyse semantic representation in grounded language learning agents that yields a plausible algorithmic account of the observed effects.

## 1 INTRODUCTION

The learning challenge faced by children acquiring their first words has long fascinated cognitive scientists and philosophers (Quine, 1960; Brown, 1973). To start making sense of language, an infant must induce structure in a constant stream of continuous visual input, slowly reconcile this structure with consistencies in the available linguistic observations, store this knowledge in memory, and apply it to inform decisions about how best to respond.

Many neural network models also overcome a learning task that is – to varying degrees – analogous to early human word learning. Image classification tasks such as the ImageNet Challenge (Deng et al., 2009) require models to induce discrete semantic classes, in many cases aligned to words, from unstructured pixel representations of large quantities of photographs (Krizhevsky et al., 2012). Visual question answering (VQA) systems (Antol et al., 2015; Xiong et al., 2016; Xu & Saenko, 2016) must reconcile raw images with (arbitrary-length) sequences of symbols, in the form of natural language questions, in order to predict lexical or phrasal answers. Recently, situated language learning agents have been developed that learn to understand sequences of linguistic symbols not only in terms of the contemporaneous raw visual input, but also in terms of past visual input and the actions required to execute an appropriate motor response (Oh et al., 2017; Chaplot et al., 2017; Hermann et al., 2017; Misra et al., 2017). The most advanced such agents learn to execute a range of phrasal and multi-task instructions, such as *find the green object in the red room*, *pick up the pencil in the third room on the right* or *go to the small green torch*, in a continous, simulated 3D world. To solve these tasks, an agent must execute sequences of hundreds of fine-grained actions, conditioned on the available sequence of language symbols and active (first-person) visual perception of the surroundings. Importantly, the knowledge acquired by such agents while mastering these tasks also permits the interpretation of familiar language in entirely novel surroundings, and the execution of novel instructions composed of combinations of familiar words (Chaplot et al., 2017; Hermann et al., 2017).

The potential impact of situated linguistic agents, VQA models and other grounded language learning systems is vast, as a basis for human users to interact with situated learning applications such as

self-driving cars and domestic robotic tools. However, our understanding of *how* these agents learn and behave is limited. The challenges of interpreting the factors or reasoning behind the decisions and predictions of neural networks are well known. Indeed, a concerted body of research in both computer vision (Zeiler & Fergus, 2014; Simonyan et al., 2014; Yosinski et al., 2015) and natural language processing (Linzen et al., 2016; Strobelt et al., 2016) has focused on addressing this uncertainty. As grounded language learning agents become more prevalent, then, understanding their learning dynamics, representation and decision-making will become increasingly important, both to inform future research and to build confidence in users who interact with such models.

We therefore aim to establish a better understanding of neural network-based models of grounded language learning, noting the parallels with research in neuroscience and psychology that aims to understand human language acquisition. Extending the approach of Ritter et al. (2017), we adapt various experimental techniques initially developed by experimental psychologists (Landau et al., 1988; Markman, 1990; Hollich et al., 2000; Colunga & Smith, 2005). In line with typical experiments on humans, our experimental simulations are conducted in a highly controlled environment: a simulated 3D world with a limited set of objects and properties, and corresponding unambiguous, symbolic linguistic stimuli (Figure 1). However, the simplicity and generality of our architecture and the form of the inputs to the model (continuous visual plus symbolic linguistic) make the proposed methods and approach directly applicable to VQA and other tasks that combine linguistic and visual data. Using these methods, we explore how the training environment of our agent affects its learning outcomes and speed, measure the generality and robustness of its understanding of certain fundamental linguistic concepts, and test for biases in the decisions it takes once trained. Further, by applying *layerwise attention*, a novel tool for visualising computation in grounded language learning models, we obtain a plausible algorithmic account of some of the effects in terms of representation and processing. Our principal findings about this canonical grounded language learning architecture are the following:

**Shape / colour biases**   When the agent is trained on an equal number of shape and colour words, it develops a propensity to extend labels for ambiguous new words according to colour rather than shape (color bias). A human-like bias towards shapes can be induced in the agent, but only if it experiences many more shape terms than colour terms during training.

**The problem of learning negation**   The agent learns to execute negated instructions, but if trained on small amounts of data it tends to represent negation in an ad hoc way that does not generalise.

**Curriculum effects for vocabulary growth**   The agent learns words more quickly if the range of words to which it is exposed is limited at first and expanded gradually as its vocabulary develops.

**Semantic processing and representation differences**   The agent learns words of different semantic classes at different speeds and represents them with features that require different degrees of visual processing depth (or abstraction) to compute.

Before describing the experiments that reveal these effects, we briefly outline details of the environment and agent used for the simulations.

## 2   A 3D WORLD FOR LANGUAGE LEARNING

Our experiments take place in the DeepMind Lab simulated world (Beattie et al., 2016), modified to include a language channel. An agent in this environment receives textual instructions, such as *find the pencil*, and is rewarded for satisfying the instruction, in this case by executing movement actions (move-left, turn right etc.) that allow it to locate a (3D, rotating) pencil and move into the space that it occupies. At each timestep in such an episode, the agent receives a continuous (RGB) pixel tensor of visual input and a symbolic (word-level) textual instruction,[1] and must execute a movement action. To solve tasks and receive rewards, the agent must therefore first learn to perceive this environment, actively controlling what it sees via movement of its head (turning actions), and to navigate its surroundings via meaningful sequences of actions.

---

[1] In all simulations in this paper, the textual instruction remains constant for the duration of each episode.

A typical simulation involves specifying certain aspects of the environment while leaving others to be determined randomly. For instance, in an object identification task, we might wish to specify the overall layout of the world, the range of positions in which objects can appear, a list of objects that can appear in each position, a probability of appearance and rewards associated with selecting each object. The environment engine is then responsible for randomly instantiating episodes that satisfy these constraints together with corresponding language instructions. Even with a detailed specification and a finite inventory of objects, properties and instruction words, there are tens of millions of unique episodes that the agent can encounter during training, each involving different object shapes, colours, patterns, shades, sizes and/or relative positions. With respect to the goal of understanding models of grounded language learning, this simulated environment and synthetic language is a useful asset: we can straightforwardly apply the methods of behavioural psychologists, testing how agents respond to precisely crafted training and test stimuli.

## 3 A SITUATED LANGUAGE LEARNING AGENT

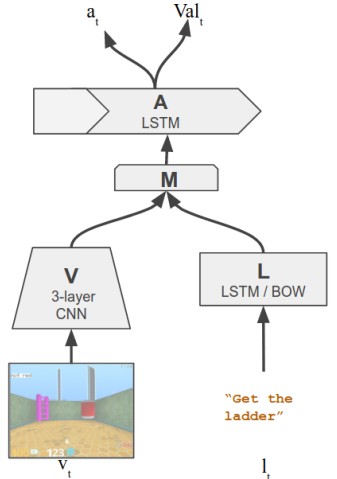
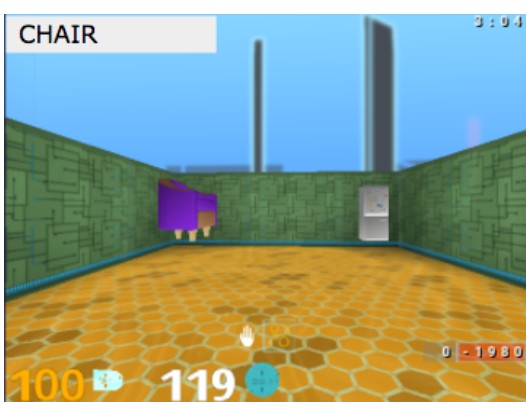

Figure 1: **Left:** Schematic agent architecture. **Right:** An example of the word learning environment common to all experiments in this paper. The agent observes two 3D rotating objects and a language instruction and must select the object that matches the instruction. In this case the instruction is a shape word (*chair*). The confounding object (a refrigerator) and the colours of both objects are selected at random and will vary across the agent's experience of the word *chair*.

For maximum generality, our simulations involve an agent that combines standard modules for processing sequential symbolic input (a recurrent network) and visual input (a convolutional network). At each time step $t$, the visual input $v_t$ is encoded by the convolutional *vision module* **V** and a recurrent (LSTM, Hochreiter & Schmidhuber (1997)) *language module* **L** encodes the instruction string $l_t$. A *mixing module* **M** determines how these signals are combined before they are passed to a LSTM *action module* **A**: here **M** is simply a feedforward linear layer operating on the concatenation of the output from **V** and **L**. The hidden state $s_t$ of **A** is fed to a policy function, which computes a probability distribution over possible motor actions $\pi(a_t|s_t)$, and a state-value function approximator $Val(s_t)$, which computes a scalar estimate of the agent value function for optimisation. *Val* estimates the expected discounted future return, by approximating the state-value function $Val_\pi(s) = \mathbb{E}_\pi[\sum_{k=0}^{\infty} \lambda^k r_{t+k+1} \mid S_t = s]$ where $S_t$ is the state of the environment at time $t$ when following policy $\pi$ and $r_t$ is the reward received following the action performed at time $t$. $0 \leq \lambda \leq 1$ represents a discount parameter. Note that this architecture is a simplified version of that proposed by Hermann et al. (2017), without auxiliary learning components.

Weight updates are computed according to the asynchronous advantage actor-critic (A3C) algorithm (Mnih et al., 2016), in conjunction with the RMSProp update rule (Tieleman & Hinton, 2012). During

training, a single parameter vector is shared across 16 CPU cores, which offers a suitable tradeoff between training time and loss of accuracy due to the asynchronous updates.

# 4  EXPERIMENTS

## 4.1  WORD LEARNING BIASES

One effect that is considered instrumental in allowing children to overcome the challenges of early word learning is the human *shape bias* (Landau et al., 1988), whereby infants tend to to presume that novel words refer to the shape of an unfamiliar object rather than, for instance, its colour, size or texture.

Our simulated environment permits the replication of the original experiment by Landau et al. (1988) designed to demonstrate the shape bias in humans. During training, the agent learns word meanings in a room containing two objects, one that matches the instruction word (positive reward) and a confounding object that does not (negative reward). Using this method, the agent is taught the meaning of a set $C$ of colour terms, $S$ of shape terms and $A$ of ambiguous terms (in the original experiment, the terms $a \in A$ were the nonsense terms 'dax' and 'riff'). The target referent for a shape term $s \in S$ can be of any colour $c \in C$ and, similarly, the target referent when learning the colours in $C$ can be of any shape. In contrast, the ambiguous terms in $A$ always correspond to objects with a specific colour $c_a \notin C$ and shape $s_a \notin S$ (e.g. 'dax' always referred to a black pencil, and neither black nor pencils were observed in any other context) . Note also that colour terms refer to a *range* of RGB space through the application of Gaussian noise to prototypical RGB codes, so that two instances of red objects will have subtly different colours.

As the agent learns, we periodically measure its bias by means of test episodes for which no learning takes place. In a test episode, the agent receives an instruction $a \in A$ ('dax') and must decide between two objects, $o_1$, whose shape is $s_a$ and whose colour is $\hat{c} \notin C \cup \{c_a\}$ (a blue pencil), and $o_2$, whose shape is $\hat{s} \notin S \cup \{s_a\}$ and whose colour is $c_a$ (a black fork). Note that in the present example neither the colour *blue* nor the shape *fork* are observed by the agent during training. As with the original human experiment, the degree of shape bias in the agent can be measured, as the agent is learning, by its propensity to select $o_1$ in preference to $o_2$. Moreover, by varying the size of sets $S$ and $C$, we can examine the effect of different training regimes on this bias exhibited by the agent.

Figure 2 illustrates how a shape/colour bias develops in agents exposed to three different training regimes. An agent that is taught exclusively colour words ($|S| = 0$, $|C| = 8$) unsurprisingly develops a strong colour bias. More interestingly, an agent that is taught an equal number of shape and colour terms ($|S| = 8$, $|C| = 8$) develops a colour bias. This suggests that the canonical architecture employed in our agent (convolutional vision network combined with language instruction embedding) naturally promotes a colour bias. In order to induce a (human-like) shape bias, it was necessary to train the agent exclusively on a larger set of ($|S| = 20$, $|C| = 0$) shapes before it began to exhibit a notable shape bias.

The fact that the network so readily develops biases that are pertinent to word learning provides insight into established effects of neural networks such as rapid acceleration of word learning (see e.g. Plunkett & Schafer (2001); Hermann et al. (2017)); it is precisely the progressive specialisation of the agent's object recognition and labelling mechanisms (towards shapes, colours or both, as determined by the training regime) that narrows the space of possible referents, permitting faster word learning as training progresses.[2]

These conclusions can be incorporated with those of Ritter et al. (2017), who observe a shape bias in convolutional networks trained on the ImageNet Challenge training set. Our experiments with a single canonical architecture exposed to different training stimuli indicate the cause of this effect to be the training data distribution (the ImageNet data indeed contains many more shape-based than colour-based categories) rather than the convolutional architecture itself. Indeed, our findings suggest that a feed-forward convolutional architecture operating (bottom-up) on image pixels promotes a colour rather than shape bias. On the other hand, a typical linguistic environment (for American children at

---

[2]We analyse this specialisation at an algorithmic level in Section 4.4.

least[3]) and, perhaps by extension, most broad-coverage machine-learning datasets, contains many more instances of shape categories than colour categories.

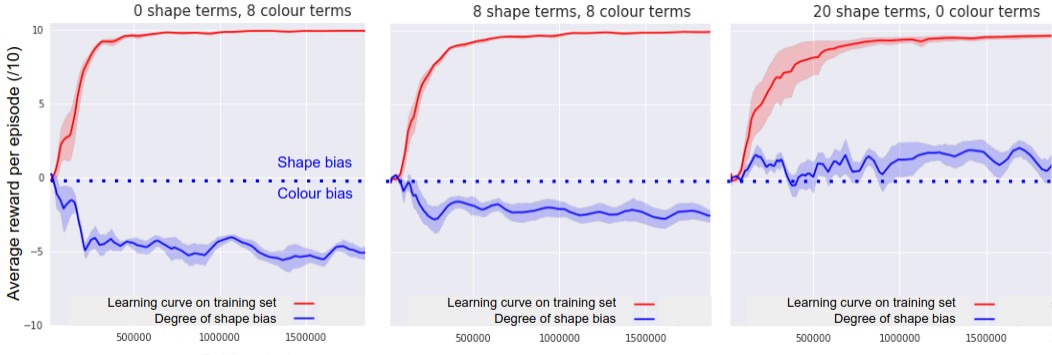

Figure 2: **Degrees of shape bias for different training regimes:** An agent that is trained only on shape words (right) more readily presumes that ambiguous words refer to object shape than to object colour. This tendency is measured across all combinations of known and confounding objects and labels and represented by the blue line. The magnitude of the bias on the scale $[-10, 10]$ is the mean 'score' (10 for the object matching the instruction in shape and 10 for the object matching in colour) over 1000 random test episodes. In contrast, an agent trained only on colour words (left) exhibits a colour bias. Interestingly, an agent trained on 8 colour and 8 shape words (middle) also exhibits a colour bias. Data (in this and proceeding figures) show mean and standard error across five fastest-learning agents of 16 different hyperparameter settings, sampled at random from ranges specified in in supplementary material 6.1.

### 4.2 THE PROBLEM OF LEARNING NEGATION

The interpretation of negated sentences such as *tell me a joke that is not offensive* is a fundamental facet of natural language understanding, and potentially critical for artificial agents receiving instructions from human users. Despite its communicative importance, negation can be challenging to acquire for human language learners. For instance, negated utterances pose greater production and comprehension difficulties than the equivalent non-negated language (Nordmeyer & Frank, 2014; Pea, 1980). To explore the acquisition of negation in grounded language learning models, we designed a simulation in which, as before, our agent was placed in a single room and required to select one of two objects matching an instruction. From a full set of training words $I$ (e.g. *red* or *ball*), a subset, $I_1 \subset I$, was sampled and presented to the agent in both positive and negative forms (*ball, not ball*) and a disjoint subset, $I_2 \subset I$, was provided only in positive forms (*pencil*). To test whether the agent could learn to understand the instruction form *pick something that is not an X* in a generally applicable way, we periodically measured its ability to interpret negated versions of the instructions in $I_2$.

As illustrated in Figure 3, the agent learned to follow both positive and negative instructions, for various sets of instruction words $I$. However, unlike other linguistic operations such as adjective-noun modification (Hermann et al., 2017), the agent exhibited difficulty generalising the notion of negation acquired in the context of $I_1$ to the held-out items $I_2$. This difficulty was most acute when $I$ consisted of 12 colour terms split evenly into $I_1$ and $I_2$. Indeed, the ability to generalise negation improved as the size of $I$ increased to include 40 shapes, from just above chance (a small positive average reward) to 75% (yielding an average reward of $\approx 5/10$). There was also a small but interesting difference in generalisation when negating shape terms vs. colour terms, which is consistent with the processing differences discussed above and in more detail in Section 4.4.

We conjecture that negation does not generalise easily because, for a word $i_n \in I$ and corresponding extension set of objects $s_n \in S$, the agent can perform perfectly on the training set by simply associating instructions of the form *'not w'* with the extension $s_1 \cup \cdots s_{n-1} \cup s_{n+1} \cdots$. This understanding of negation would generalise much worse than an interpretation of *'not w'* that

---

[3]As verified by simple analysis of the child-directed language corpus Wordbank (Frank et al., 2017).

involved identifying and avoiding an object of type *w*. For small training sets, the results suggest that the model prefers the former interpretation, but also that its tendency to discover the latter more generalisable understanding increases as the set of negated concepts to which it is exposed grows. Thus, with appropriately broad exposure to instances of negation during training, neural networks without bespoke biases or regularization can learn to respond effectively to negated stimuli pertaining to their perceptible surroundings. However, tailored architectures and computational biases may be required in cases where agents learn from, and act on, constrained sets of linguistic stimuli.

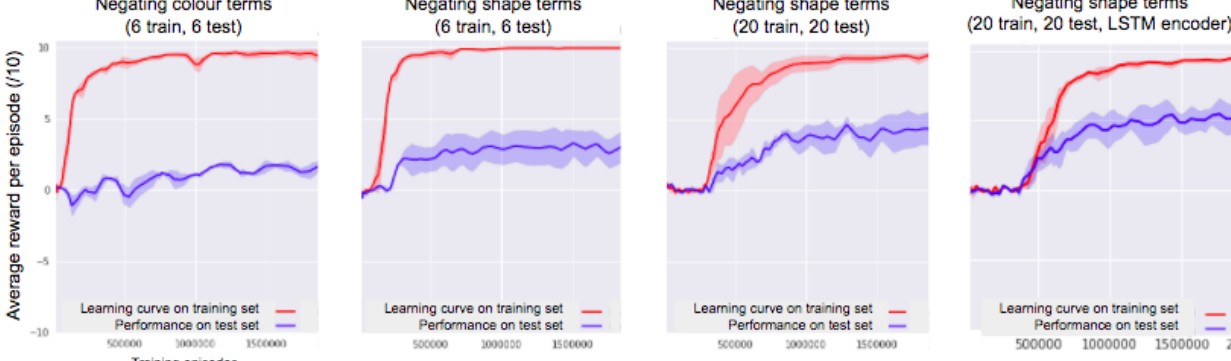

Figure 3: **The problem of learning negation in language learning agents:** The agent must be exposed to negative instructions in a sufficiently diverse range of contexts in order to learn a useful generalisable notion of negation. If trained to interpret positive commands involving 12 terms and negative commands involving 6 of those 12 terms (left, colour terms, middle, shape terms), the agent does not effectively interpret negative commands involving the remaining 6 terms. When exposed to 40 shape terms and trained to interpret negative commands involving 20 of those terms, the agent generalises the negation operation more effectively, but still not perfectly. When the two-word negative instructions are encoded with an LSTM rather than additive BOW encoder, an almost identical pattern of gradually improving generalisation is observed.

## 4.3 CURRICULUM LEARNING

The idea that learning is more successful if simpler things are studied before more complex things is a basic tenet of human education. There is some consensus that early exposure to simple, clear linguistic input helps child language acquisition (Fernald et al., 2010), although this is not unanimous (Shore, 1997). Controlled experiments with artificial neural networks trained directly on symbolic (language-like) data have also revealed faster or more effective learning when training examples are ordered by some metric of complexity (Elman, 1993). This approach is now typically referred to as curriculum learning (Bengio et al., 2009). However, robust improvements due to curriculum learning can be difficult to achieve in the context of text-based learning (Mikolov et al., 2011; Graves et al., 2017), and curricula are not ordinarily applied when training text-based neural language models. Recent evidence suggests that the benefits of curriculum training can be more easily realised for agents learning to act conditioned on language than those learning to map between linguistic inputs and outputs. Both Hermann et al. (2017) and Oh et al. (2017) observed that curricula were essential for agents learning to execute linguistic instructions that require both resolving of referring expressions (*get the red ball..*) and non-trivial action policies such as exploration (*..in the green room*).

Here, we chose to explore curriculum learning in a more controlled way in grounded language learning agents. To do so, we trained our agent to learn the meaning of 40 shape words (as exhibited by its ability to respond appropriately) under two conditions. In one condition, the agent was presented with the 40 words (together with corresponding target and confounding objects) sampled randomly throughout training. In another condition, the agent was only presented with a subset of the 40 words (selected at random) until these were mastered (as indicated by an average reward of $9.8/10$ over 1000 consecutive trials), at which point this subset was expanded to include more words. This process was iterated for subsets of size 2, 5, 10 and eventually 40 words.

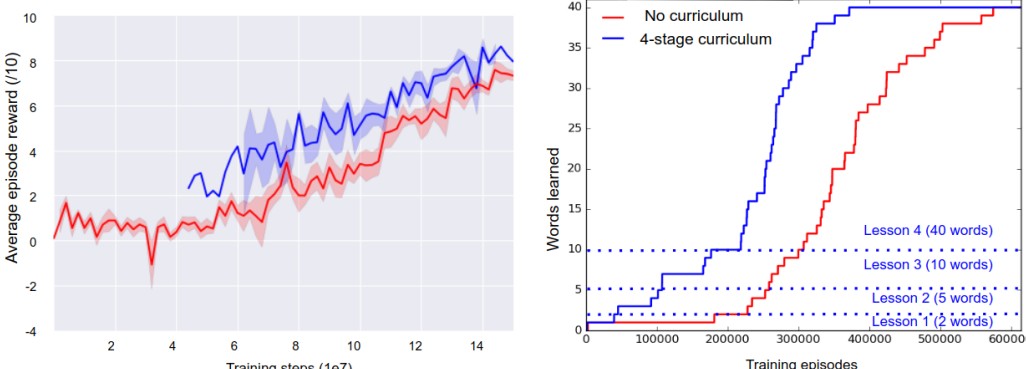

Figure 4: **Curriculum training expedites vocabulary growth:** An agent that is presented with stimuli sampled uniformly from a set $S$ of 40 shape words (red line) learns more slowly than one whose stimuli are constrained to a two-word subset $S_1, S_1 \subset S$, until the agent learns both words, then extended to a 5-word subset $S_2, S_1 \subset S_2 \subset S$, then a 10-word subset $S_3, S_2 \subset S_3 \subset S$. This strong effect of 'curriculum learning' can be observed both when comparing average reward when agents in the two conditions are learning words sampled from $S$ (left - note that the agent in the curriculum condition begins reporting performance on $S$ after prior training on the restricted subsets) and by measuring vocabulary size as a function of training episodes (right).[5]

As shown in Figure 4, an agent that followed the curriculum (i.e. in the second condition) learned notably faster overall than one presented directly with a large group of new words. This result further corroborates the importance of training curricula for grounded language learning agents. Moreover, unlike the effect observed by Hermann et al. (2017), which focused on tasks requiring the agent to explore a large maze, the present simulation demonstrates strong curriculum effects simply when learning to associate objects with words. Thus, the development of core linguistic and semantic knowledge in situated or grounded agents can be clearly expedited by starting small and easy and slowly increasing the language learning challenge faced by the agent.

## 4.4 PROCESSING AND REPRESENTATION DIFFERENCES

Many studies aiming to understand linguistic processing in humans do so by uncovering connections between different semantic or conceptual domains and distinct processing or patterns of representation. These effects emerge via both behavioural methods (measuring differences in how subjects learn, use or even forget concepts of different types (Bowerman & Choi, 2001; Paivio et al., 1994)) and neuroimaging (associating words of different types with spatially and structurally distinct brain regions (Huth et al., 2016; Patterson et al., 2007)). Neuroscientific theories of memory, representation and learning have been developed to account for these effects (Rogers & McClelland, 2004; Binder & Desai, 2011). In the pursuit of a better understanding of artificial agents, we can similarly explore links between word classes, semantic domains and patterns of learning, behaviour, processing or representation. This knowledge could ultimately be essential for informing the process of designing architectures capable of learning not just the simple language studied here, but also the full range of abstract semantic phenomena inherent in adult language.

**Word learning speeds** The order in which words of different types are acquired has been used to inform theories of child language acquisition (Gentner, 1982) and human semantic processing more generally (Ghyselinck et al., 2004). To explore the order of word learning in our agent, we exposed it to randomly interleaved training instances for words of six different classes (shapes, colours, patterns, sizes, shades and superordinate category terms, such as *furniture*), for multiple shapes. We compared the rates of word learning in two conditions. In the *fixed class-size*, each class was restricted to two exemplars. In the *variable class-size* condition, each class was represented by a different number of

---

[5]Here, a word is considered to be 'known', and thus part of the agent's vocabulary, if the agent responds correctly to that word for 50 consecutive exposures.

members, a more faithful reflection of natural language, where one word class (e.g. prepositions) can have a different number of members from another (e.g. nouns).[6] In both conditions, the training stimuli were sampled random uniformly from all word types (not word classes), so that an agent in the variable class-size condition received approximately four times as much exposure to shape words as to colour words, for instance.

As illustrated in Figure 5, there were clear differences in the speed with which the agent learned words of different classes. In the fixed class-size condition, the first words to be learned were *blue* (a colour word) and *diagonal-striped* (a pattern), with the second colour word, *brown*, learned around the same time as the two shapes *chair* and *suitcase* and the relative size terms *larger* and *smaller*. Category terms were learned after shape terms.[7] In contrast, in the variable class-size condition the variable exposure to different word classes seems to cause a degree of specialisation in shape words, so that the agent learns all 40 shape words well before it acquires the 12 colour words.

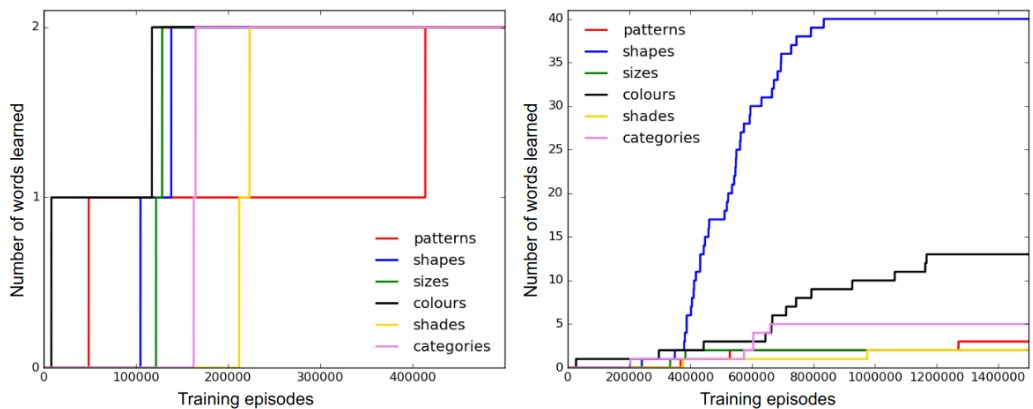

Figure 5: **Words from different semantic classes are learned at different speeds:** In the fixed class-size condition (left), the agent learns two words from each class. In the variable class-size condition (right) each class has a different number of members, as per supplementary material 6.2.

**Layer-wise attention**    To complement our behavioural analysis we developed a method for better understanding semantic processing and representation in the agent. The method, which we call *layer-wise attention*, involves modifying the agent architecture to expose processing differences in the visual features that are most pertinent to each lexical concept. In the standard agent, a distributed representation of the linguistic input is concatenated with the output from the top layer of a 3-layer convolutional visual module at each timestep, fed through a multi-layer perceptron and then passed to the agent's (recurrent) core. We modify this agent so that it can learn to attend to the output from different layers of its visual processing module, conditioned on the linguistic input available at a particular moment.

Let $e_l$ be the representation of a language instruction $l$ and $\mathbf{v}_i$ be the output of layer $i = 1, 2, 3$ of the visual module with dimension $n_i \times n_i \times k_i$, where $k_i$ is the number of feature maps. In the layerwise attention module, the $\mathbf{v}_i$ are first passed through 3 independent linear layers to $\mathbf{v}'_i$ with common final dimension $n_i \times n_i \times K$, such that $K$ is also the dimensionality of $e_l$. The $\mathbf{v}'_i$ are then stacked into a single tensor $T$ of dimension $d \times K$, where $d = \sum_{i=1}^{3} n_i^2$. $T$ is then multiplied by $e_l$ and passed through a softmax layer to yield a $d$ dimensional discrete probability distribution over all (pixel-like) locations represented in each layer of the visual module $\mathbf{V}$. These values are applied multiplicatively to each of the ($k_i$-dimension) representations returned by $\mathbf{V}$, before a pooling step (mirroring that of the final layer in the original agent) and then concatenation.

Layerwise attention provides a measure not only of which image locations contain the most important information for the agent when choosing actions at a given timestep, but also at what level of (visual) abstraction that information is most useful. This insight can be visualised by applying the method of Simonyan et al. (2014), propagating the probability mass from the attention distribution back

---

[6]See supplementary material 6.2 for details.

[7]Humans also learn basic level categories before superordinate category terms (Horton & Markman, 1980).

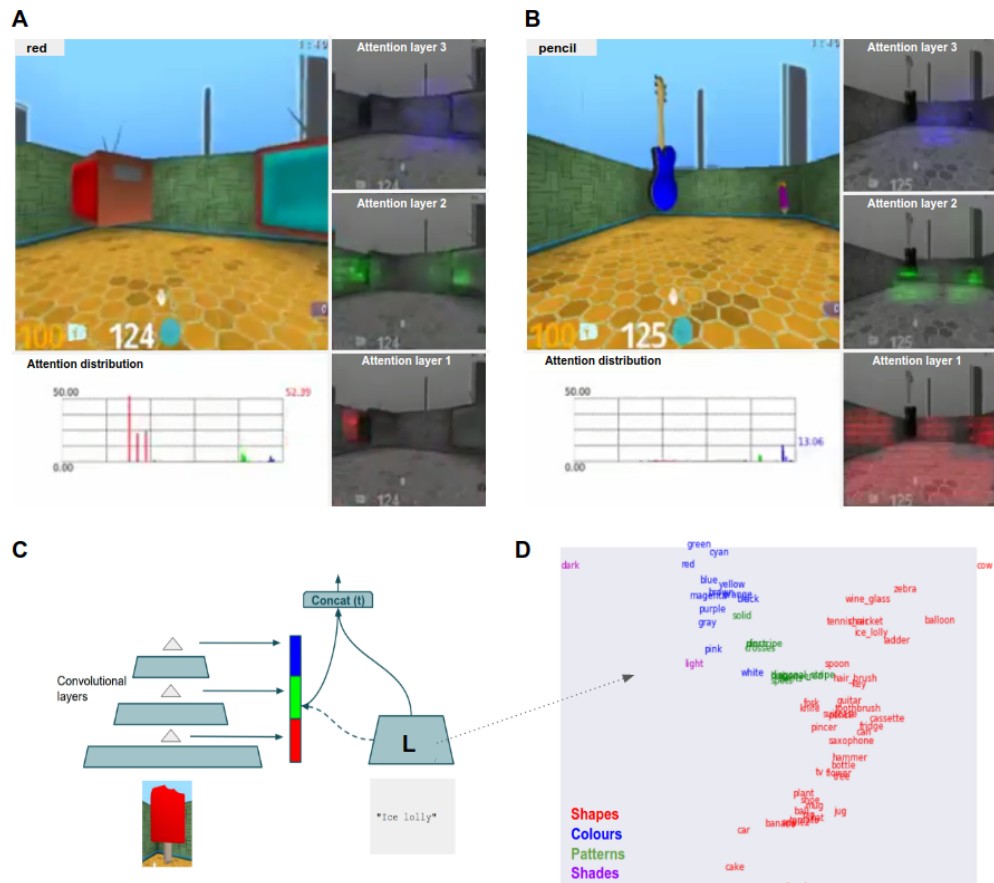

Figure 6: **Representation and processing differences between colour and shape words.** 'Dashboards' for interpreting processing in an agent with layerwise attention. The large pane at the top left of the dashboard shows the input to the agent. The bar chart on the bottom left shows the attention distribution over all 520 'locations' from the agent's visual representations. 400 red bars show the attention on the $(20 \times 20)$ locations output from the lowest layer of the convnet, 81 green bars show the attention on the $(9 \times 9)$ locations from the middle layer and 49 blue bars show the attention on the $(7 \times 7)$ locations from the top layer. The small windows on the right side illustrate these attention weights (grouped by layer) propagated back to and superimposed over a greyscale copy of the input image, as described by Simonyan et al. (2014). An agent trained exclusively on colour words (**A**) relies more on the first and second layers of the convnet than an agent trained exclusively on shape words (**B**), which uses second and upper layer visual features. **C**: A schematic of layerwise attention in the agent architecture. **D**: A 2D (t-SNE) visualisation of the space of the word embeddings weights in the language module (**L**) of an agent trained on different word types, illustrating that words cluster naturally according to semantic classes in the linguistic memory of the agent.

onto the input image. Figure 6 illustrates the effect of backpropagating the attention probabilities corresponding to each layer of the convnet onto (grayscale copies of) the visual input.[8] As is clear from these visualisations, an agent that is exposed only to shape words will learn to rely on features from the upper-most layers of its visual module when considering objects in its surroundings. In contrast, an agent trained to interpret only colour terms focuses with feature detectors from the lower layers of its visual module in order to distinguish between objects of interest. It is well established that convolutional networks trained to classify images also exhibit differential specialisation of feature detectors between layers (see e.g. LeCun et al. (2010)). Layerwise attention provides a means to

---

[8]Best viewed in video format at `https://drive.google.com/file/d/0B2RnkWT-MR0TbHRrUktkQUpBMzg/view?usp=sharing`.

quantify the magnitude of this specialisation, and to measure the importance of each layer with respect to particular linguistic stimuli. It is also notable that a more conventional 2D (T-SNE) visualisation of the word embeddings in the input layer of **L** provides further evidence of of word-class-specific processing, as illustrated in Figure 6.

## 5 CONCLUSION

Models that are capable of grounded language learning promise to significantly advance the ways in which humans and intelligent technology can interact. In this study, we have explored how a situated language learning agent built from canonical neural-network components overcomes the challenge of early language learning. We measured the behaviour exhibited once the first words and simple phrases are acquired, tested factors that speed up this learning, explored aspects of language that pose particular problems and presented a technique, layerwise attention, for better understanding semantic and visual processing in such agents.

The application of experimental paradigms from cognitive psychology to better understand deep neural nets was proposed by Ritter et al. (2017), who observed that convolutional architectures exhibit a shape bias when trained on the ImageNet Challenge data. The ability to control precisely both training and test stimuli in our simulated environment allowed us to isolate this effect as deriving from the training data, and indeed to reach the opposite conclusion about the architecture itself. This study also goes beyond that of Ritter et al. (2017) in exploring more abstract linguistic operations (negation, abstract category terms) and studying curriculum effects on the dynamics of word learning. Further, we complement these behavioural observations with computatoinal analysis of representation and processing, via layerwise attention.

While the control and precision afforded by the simulated environment in the present study has made these analyses and conclusions possible, in future, as our understanding of language learning agents develops, it will be essential to verify conclusions on agents trained on more naturalistic data. At first, this might involve curated sets of images, videos and naturally-occurring text etc, and, ultimately, experiments on robots trained to communicate about perceptible surroundings with human interlocutors. In a world with agents capable of learning such advanced linguistic behaviour, it would certainly be more challenging, but also even more crucial, to understand not just what they can do, but also how they learn to do it.

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

## 6 SUPPLEMENTARY MATERIAL

### 6.1 AGENT DETAILS

| Hyperparameter | Value | Description |
|---|---|---|
| train_steps | 640m | Theoretical maximum number of time steps (across all episodes) for which the agent will be trained. |
| env_steps_per_core_step | 4 | Number of time steps between each action decision (action smoothing) |
| num_workers | 32 | Number of independent workers running replicas of the environment with asynchronous updating. |
| unroll_length | 50 | Number of time steps through which error is backpropagated in the core LSTM action module |
| **visual encoder** | | |
| $num_layers$ | 3 | Layers in the convolutional vision network. |
| $output_channels$ | (32, 64, 64) | Number of feature maps in each layer of the network. |
| $kernel_shapes$ | (8, 4, 3) | Shapes of the (square) convolutional kernels in each layer of the network. |
| strides | (4, 2, 1) | Convolution stride length in each layer of the network. |
| activation | relu | Activation function applied after all except the final layer of the visual encoder. |
| **language encoder** | | |
| encoder_type | BOW | Whether the language encoder uses an additive bag-of-words (BOW) or an LSTM architecture (with tanh nonlinearity). |
| $embedding_dim$ | 128 | Dimension of the word and instruction embeddings. |
| **cost calculation** | | |
| additional_discounting | 0.99 | Discount used to compute the long-term return R_t in the A3C objective |
| cost_base | 0.5 | Multiplicative scaling of all computed gradients on the backward pass in the network |
| **optimisation** | | |
| clip_grad_norm | 100 | Limit on the norm of the gradient across all agent network parameters (if above, scale down) |
| decay | 0.99 | Decay term in RMSprop gradient averaging function |
| epsilon | 0.1 | Epsilon term in RMSprop gradient averaging function |
| learning_rate_finish | 0 | Learning rate at the end of training, based on which linear annealing of is applied. |
| momentum | 0 | Momentum parameter in RMSprop gradient averaging function |

Table 1: Agent hyperparameters that are fixed throughout our experimentation but otherwise not specified in the text.

| Hyperparameter | Value | Description |
|---|---|---|
| **language encoder** | | |
| embed_init | *uniform(0.5, 1)* | Standard deviation of normal distribution (mean = 0) for sampling initial values of word-embedding weights in **L.** |
| **optimisation** | | |
| entropy_cost | *uniform(0.0005, 0.005)* | Strength of the (additive) entropy regularisation term in the A3C cost function. |
| learning_rate_start | *loguniform(0.0001, 0.002)* | Learning rate at the beginning of training annealed linearly to reach learning_rate_finish at the end of train_steps. |

Table 2: Agent hyperparameters that randomly sampled in order to yield different replicas of our agents for training. *uniform(x, y)* indicates that values are sampled uniformly from the range $[x, y]$. *loguniform(x, y)* indicates that values are sampled from a uniform distribution in log-space (favouring lower values) on the range $[x, y]$.

### 6.2 EXPERIMENT DETAILS

| Word class | Words |
|---|---|
| Shapes (40) | *chair, suitcase, tv, ball, balloon, cow[1], zebra[1] cake, can[5], cassette, chair, guitar, hair-brush[2], hat[4], ice-lolly, ladder, mug[5], pencil[2], suitcase, toothbrush[2], key[2], bottle[5], car, cherries[3], fork[2], fridge, hammer[2], knife[2], spoon[2], apple[3], banana[3] flower, jug[5], pig[1], pincer[2], plant[3], saxophone, shoe[4], tennis-racket, tomato[1], tree[3], wine-glass[5]* |
| Colours (12) | *blue, brown, pink, yellow, red, green, cyan, magenta, white, grey, purple* |
| Categories (5) | *1:animals, 2:tools, 3:plants, 4:clothing, 5:containers* |
| Patterns (3) | *diagonal-striped, chequered, spotted* |
| Shades (2) | *lighter, darker* |
| Sizes (2) | *larger, smaller* |

Table 3: Word classes (class size) used in the word learning speed experiment in Section 4.4. Superscript indicates if the shape a word refers to is also in the extension of a category word.

