# OpenReview forum: "Understanding Grounded Language Learning Agents"
_ICLR.cc/2018/Conference — Reject_

### Official Review · AnonReviewer2 · 2017-11-12
**Worthy goal, but implementation felt a bit underwhelming**

**Rating:** 4
**Confidence:** 5

**Review:**

This paper presents an analysis of an agent trained to follow linguistic commands in a 3D environment.  The behaviour of the agent is analyzed by means of a set of "psycholinguistic" experiments probing what it learned, and by inspection of its visual component through an attentional mechanism.

On the positive side, it is nice to read a paper that focuses on understanding what an agent is learning. On the negative side, I did not get many new insights from the analyses presented in the study.

3 A situated language learning agent

I can't make up the chair from the refrigerator in the figure.

4.1 Word learning biases

This experiment shows that, when an agent is trained on shapes only, it will exhibit a shape bias when tested on new shapes and colors. Conversely, when it is exposed to colors only, it will have a color bias. When the training set is balanced, the agent shows a mild bias for the simpler color property. How is this interesting or surprising? The crucial question, here, would be whether, when an agent is trained in a naturalistic environment (i.e., where distributions of colors, shapes and other properties reflect those encountered by biological agents), it would show a human-like shape bias. This, however, is not addressed in the paper.

Minor comments about this section:

- Was there noise also in shape generation, or were all object instances identical?

- propensity to select o_2: rather o_1?

- I did not follow the paragraph starting with "This effect provides".

4.2 The problem of learning negation

I found this experiment very interesting.

Perhaps, the authors could be more explicit about the usage of negation here. The meaning of commands containing negation are, I think, conjunctions of the form "pick something and do not pick X" (as opposed to the more natural "do not pick X").

modifiation: modification

4.3 Curriculum learning

Perhaps the difference in curriculum effectiveness in language modeling vs grounded language learning simulations is due to the fact that the former operates on large amounts of natural data, where it's hard to define the curriculum, while the latter are typically grounded in toy worlds with a controlled language, where it's easier to construct the curriculum.

4.4 Processing and representation differences

There is virtually no discussion of what makes the naturalistic setup naturalistic, and thus it's not clear which conclusions we should derive from the corresponding experiments. Also, I don't see what we should learn from Figure 5 (besides the fact that in the controlled condition shapes are easier than categories). For the naturalistic condition, the current figure is misleading, since different classes contain different numbers of instances. It would be better to report proportions.

Concerning the attention analysis, it seems to me that all it's saying is that lower layers of a CNN detect lower-level properties such as colors, higher layers detect more complex properties, such as shapes characterizing objects. What is novel here?

Also, since introducing attention changes the architecture, shouldn't the paper report the learning behaviour of the attention-augmented network?

The explanation of the attention mechanism is dense, and perhaps could be aided by a diagram (in the supplementary materials?). I think the description uses "length" when "dimensional(ity)" is meant.

6. Supplementary material

It would be good to have an explicit description of the architecture, including number of layers of the various components, structure of the CNN, non-linearities, dimensionality of the layers, etc. (some of this information is inconsistently provided in the paper).

It's interesting that the encoder is actually a BOW model. This should be discussed in the paper, as it raises concerns about the linguistic interest of the controlled language that was used.

Table 3: indicates is: indicates if

---

> ### Author Response · Authors · 2017-12-29
> **We explain clearly why we find the results interesting and surprising. Could you explain more objectively why they are uninteresting?**
>
> Thank you. We have responded to the major points, minor corrections are fixed in the text.
>
> Concern: I did not get many new insights from the analyses presented in the study.
> [shape/colour bias] How is this interesting or surprising?
>
> Response: We obtained many new insights when doing the research. If you cite where you learned about the effects we report we will cite that work and position our work relative to it.
>
> We find the bias results both interesting and surprising given previous work. Note we are not just saying 'the trained model behaves differently depending on the training data'. We claim that convnet+lang embedding models exhibit a strong colour bias when extending ambiguous labels (assuming unbiased training data). We have made this conclusion more explicit in the paper. It surprises us because (a) it opposes an established effect in humans and (b) Ritter et al. showed similar models trained on imagenet photos exhibit the opposite bias. Unlike them, we isolate the cause of the bias (the architecture), by controlling for bias in the training data. This is relevant to current research because most approaches to grounded language learning involve a related architecture.
>
>
> Concern: The crucial question ... when an agent is trained in a naturalistic environment..., it would show a human-like shape bias.
>
> Response: This question is not crucial to our goal, which is to understand artificial models, not humans. Any researcher would love to experiment with agents in a perceptual environment that reflects a 'typical' human-machine interaction. However, as far as we know (please say if you disagree), nobody has done this, and it would be very challenging. Where would the model be? Would it learn from expert humans, novice humans, other agents? Rather than try to estimate these unknowns, we studied differences when changing unambiguous factors in the environment (e.g. shape /colour words ratio, equal-sized vs. variable-sized word classes). We make conclusions about how the model's learning is affected by these clear differences in experience, which will be applicable once such models can interact with 'real' users in the real world.
>
> Concern: Concerning the attention analysis..... What is novel here?
>
> Response: You don't cite why this is not novel. Are you are thinking of research into the feature-maps learned at different layers of convnet image classifiers? The novelty over that - we propose a method to visualise and quantify the interaction between language and vision as word meanings are combined and composed (and as a trained agent explores and acts in the world). Using this method, we can see what visual information is most pertinent to the meaning of any linguistic stimuli, including novel phrases not seen during training. This is certainly different from conclusions about how visual classifiers. The fact that our findings (about a network that can learn to combine language and vision at different levels of abstraction in order to solve tasks) are consistent with those findings (about a network trained to classify images by predicting a label) does not render either result redundant.
>
> Concern: There is no discussion of what makes the naturalistic setup naturalistic
>
> Response: Naturalistic - not all word classes have the same number of members like e.g. the class of prepositions has many fewer members than the class of nouns. We have changed the term as it was misleading.
>
> Concern: The encoder is actually a BOW model...raises concerns about the linguistic interest of the controlled language that was used.
>
> Response: Good point. In most experiments the input is a single word, so BOW and LSTM are equivalent.. The exception is negation experiments, which we have repeated with both BOW and LSTM encoding, and report the results in Figure 3. The effect remains.
>
> Concern: Perhaps difference in curriculum effectiveness.....
> Response: We don't claim that curriculum learning never works for text-based learning, only that it was easy to observe strong curriculum effects in the context of our simulated world and situated agent. We have changed the text to make this more precise.
>
> Concern: shouldn't the paper report the learning behaviour of the attention-augmented network?
>
> Response: We did not notice learning differences (for instance in sample-complexity), but layerwise attention needs additional computation which makes clock time slower. We did not experiment with this attention more generally since it would have made the findings less general. We now explain this reasoning.
>
> Question: Was there noise also in shape generation, or were all object instances identical?
> All objects were identical in size but rotate to give variation in perspective. We have added this detail.
>
> Other requested improvements:
> Re-worded the paragraph beginning "This effect provides"
>
> More explicit about the nature of the negation command
>
> Added full details of the model in supplementary material.

---

> > ### Comment · AnonReviewer2 · 2017-12-31
> > **Thanks for the reply**
> >
> > I thank the authors for their thorough and thoughtful response, and for incorporating my feedback in the paper revision.
> >
> > I still don't grasp the main point of the experiment on learning biases. I agree that it makes sense to study learning in an artificial, controlled setup. But, when this setup involves feeding the agent more color or more shape terms, the result that the model will show a color/shape bias seems trivial to me. This leaves, as the only interesting find, the condition in which the model is getting the same amount of color and shape examples, and it displays a (mild) colour bias. This might be surprising, but then the authors should provide some insight, possibly by means of follow-up experiments, on *why* such bias should emerge. Your tightly controlled setup and relatively simple architecture should allow you to gain a fuller understanding of such biases. If not, then I don't see the point of using an artificial simulation.
> >
> > Concerning the attention analysis, I was indeed thinking of research about what convnets learn at different layers, and it is true that you went beyond that by looking at which components are activated during word learning. However, you illustrate your mechanism when it is applied to very basic color and shape words, that are not that different from attribute labels of the sort routinely used in computer vision. As an example of something I'd find exciting, it would be cool if you could show that, say, the agent pays more attention to the color-encoding layers for fruits (for which colour is often a distinctive property) than for tools (for which it typically isn't).
> >
> > Let me re-iterate that I really like the emphasis of the paper on analyzing the behaviour of a learning agent, rather than on quantitative performance. However, except for the negation study, I did not learn much from the paper. I would be hard-pressed to tell what the general take-home lessons (either for AI or cognitive science) are. To conclude, I think that this is a very promising research line, that is however not yet ripe for publication in a major conference.

---

> > > ### Author Response · Authors · 2017-12-31
> > > **This paper contains more and wider contributions than various previously published and important papers**
> > >
> > > Thanks for your response. It made your position clearer to us, which was really helpful, and we of course appreciate and respect your views. However, we want to make clear that we entirely disagree with them, particularly your suggestion that the work is somewhat half-baked and your assessment of the amount of insight and contributions in the paper.
> > >
> > > To re-iterate, the take-home lessons of the paper are:
> > >
> > > 1. architectures that combine convnet vision with (language-like) sequences of words, and use the aquired semantic representations to solve tasks, naturally promote a bias to extend ambiguous new labels according to colour, not shape.
> > > 2. when such architectures are trained on a lot of shape words, they can exhibit the opposite (human-like) bias.
> > > 3. although we are not sure exactly why these effects emerge, we provide some insight by demonstrating that representation (and decisions about) colour words involves focusing on information that is most easily extracted at the lower-level of visual processing, whereas processing shape words requires focus on the higher-levels of visual processing
> > > 4. when learning to execute instructions involving a form of negation, such architectures typically learn to generalise in an 'inefficient' non-generalisable way unless the training experience is sufficiently broad, in which case this limitation can be resolved to some degree.
> > > 5. when learning multiple words, such architectures learn much faster if their experience is restricted initially to some of the words, and only broadened once the initial words are mastered
> > >
> > > You may consider 1-5 to be insufficient for a conference publication in this field. We believe this view is at odds with the following facts (among others):
> > >
> > > - A study that focused on (1-2) - demonstrating a shape-bias in visual-classification architectures when trained on imagenet images  - was published at last year at ICML (https://arxiv.org/pdf/1706.08606.pdf).
> > > - This paper, raised by Reviewer 1, was (https://pdfs.semanticscholar.org/371b/240bebcaa68921aa87db4cd3a5d4e2a3a36b.pdf) was published in the journal Cognitive Science and observed an effect similar to that of (4), but when learning from synthetic symbolic (sequential) data.
> > > - The well-cited and influential paper published at ICML (https://qmro.qmul.ac.uk/xmlui/bitstream/handle/123456789/15972/Bengio%2C%202009%20Curriculum%20Learning.pdf?sequence=1) focused solely on demonstrating that neural nets (learning from synthetic symbolic or pixel data or natural language text) tend to learn faster if the inputs are ordered according to difficulty in some way (5).
> > >
> > > The above papers each highlighted what have now become (or are becoming) commonly-accepted effects of learning in neural networks. However, each paper focused only on a subset of the phenomena we study in this work. Further, none of the above papers was able to explain *why* effects emerge, beyond offering some rudimentary analyses and intuition. Nonetheless, we think they are all excellent contributions to the understanding of neural networks. We don't want to suggest that our work is superior to these, only that the scope of insight and novel findings covered by our paper seems to be at least on-par with these works, each of which was deemed publishable by the community.
> > >
> > >  We agree that studying the *why* question around each of these effects is very important as a follow-up, but it is an extremely difficult goal, and an objective that will probably be reached very gradually by the whole community working together (building on contributions like ours). To require such a fundamental breakthrough for a conference publication (rather than clear evidence that we have moved knowledge forward, like we provide) is, in our opinion, very harsh. Moreover, even if we just did a few more experiments in the *why* direction for one of 1-5, something would need to be omitted, since (as Reviewer 1 has raised) we are already at the page limit. We feel that would detract from the paper in other ways. This is also the case for the follow-up experiments that you suggest for the attention analysis. We agree that these would be really interesting, and we are glad that you can now see the potential of using layerwise attention to understand linguistic representations. In conclusion, we believe that the current scope and content of the paper includes more than enough new insights and takeaways for a self-contained contribution on understanding grounded language learning with neural networks.

---

> > > > ### Comment · AnonReviewer2 · 2018-01-02
> > > > **wider is not necessarily better**
> > > >
> > > > Thanks for your further response. To clarify once more, I find your research direction very exciting. Also, I am sorry if I implied that your work is "half-baked": that was not my intention. I can see that it's very thorough. It just seems to me that, with the exception of the negation study, it is not yet providing novel results of some generality, or addressing *why* questions in a way that would be helpful to the community.
> > > >
> > > > Elman's classic paper provided an insight that was completely novel at the time about how the order of presentation of input data affects RNN learning of natural linguistic structures, and a thorough qualitative analysis of the networks' learning behaviour.  Bengio et al. introduced the idea of curriculum learning to the ML community. I was not familiar with the work of Ritter et al. before reading your paper, but, based on a cursory look at it, it seems to me it presents a comparison between learning concepts from a controlled data-set and in the wild. I am sorry to be stubborn, but I found these 3 studies more instructive than yours, in its current version.
> > > >
> > > > I realize, of course, that you can't cover everything in a conference paper. Indeed, it would be great to see separate papers on the various topics you are presenting, with more detailed analyses of each.

---

### Official Review · AnonReviewer1 · 2017-11-27
**An innovating and interesting approach, but there are serious, lingering concerns about the approach**

**Rating:** 5
**Confidence:** 3

**Review:**

In this manuscript, the authors connect psychological experimental methods to understand how the black box of the mind solves problems with current issues in understanding how the black box of deep learning methods solves problems. The authors used situated versions of human language learning tasks as simulation environments to test a CNN + LSTM deep learning network. They examined a few key phenomena: shape/color bias, learning negation concepts, incremental learning, and how learning affects the representation of objects via attention-like processes. They illustrated conditions in which their deep learning network acts similarly to people in simulations.
Developing methods that enable humans to understand how deep learning models solve problems is an important problem for many reasons (e.g., usability of models for science, ethical concerns) that has captured the interest of a wide range of researchers. By adapting experimental methodology from psychology to test that have been used to understand and explain the internal workings of the mind, the authors approach the problem in a novel and innovative manner. I was impressed by the range of phenomena they tackled and their analyses were informative in understanding the behavior of deep learning models
I found the analogy persuasive in theory, but I was not convinced that the current manuscript really demonstrates its value. In particular, I did not see the value of situating their model in a grounded environment. One analysis that would have helped convince me is a comparison to an equivalent non-grounded deep learning model (e.g., a CNN trained to make equivalent classifications), and show how this would not help us understand human behavior. However, the more I thought about the logic of this type of analysis, the more concerned I became about the logic of their approach.
What would it mean if the equivalent non-situated model does not show the phenomena? If it does not, it could illustrate the efficacy of using situated environments. But, it also could mean that their technique acts differently for equivalent situated and non-situated models. In this case though, what would we learn about the more general non-situated case then? It does not seem like we would learn much, which would defeat the purpose of the technique. Alternatively, if the equivalent non-situated model does show the phenomena, then using the situated version would not be useful because the model acts equivalently in both cases. I am not fully convinced by the argument I just sketched, but leaves me very concerned about the usefulness of their approach. (note that the “controlled” vs. “naturalistic” analyses in the word learning section did not convince me. This argues for the importance of using naturalistic statistics – not necessarily cross-modal, situated environments as the authors argue for).
Additionally, I was unconvinced that simpler models could not be used to examine the phenomena that they analyzed. Although combining LSTM with CNN via a “mixing” module was interesting, it added another layer of complexity that made it more difficult to assess what the results meant. This left me less convinced of the usefulness of their paradigm. If we need to create a novel deep learning method to illustrate its efficacy, how will it be useful for solving the problem that motivated everything: understanding how pre-existing deep learning methods solve problems.

---

> ### Author Response · Authors · 2017-12-29
> **Very controlled and naturalistic tasks are both critical for understanding deep learning and RL**
>
> Thanks for your review, we appreciate your effort in considering the paper. You raised some very interesting concerns and questions that we have thought about a lot during the course of conducting this research. If we understand correctly, your greatest worry is with our use of a simulated learning environment, and the feeling that our results may not generalise to models that learn from other types of data.
>
> Our response:
>
> To address the goals of the paper (understanding how randomly-initialised neural nets can learn 'language' from raw, unstructured visual and linguistic input), we needed to decide on stimuli for training and testing. When selecting stimuli for understanding learning machines, there is necessarily a trade-off between the realism and control. Most studies on human learning (in neuroscience, psychology etc) use a small set of controlled stimuli (for instance, photographs or pictures of objects on a table). These stimuli have much less variation in lighting, angles, colours etc. that the real world, but this lack of variation makes it easier to understand the connection between the factors that do vary in the stimuli and the behaviour of the learner. Such control makes more precise measurement, comparison, replication and, in many instances, conclusion possible. When experimenting with artificial learning machines, there is similarly advantages and disadvantages to experimenting with more controlled vs more realistic stimuli. If we had chosen to experiment with sets of photographs (such as those in the ImageNet challenge), each individual data point would have constituted a more realistic test of visual processing, but we would also have introduced a host of confounds and challenges are absent in the highly-controlled, synthetic world. For instance, it would not have been possible to change the colour of an object while keeping its shape the same, or to study curriculum effects or negation independently of the specifics of the particular images involved.
>
> We believe strongly that a complete understanding of the dynamics of learning and representation in neural nets will require both studies on noisy, naturalistic and work with controlled, synthetic stimuli. Indeed, to date, many of the most important exploratory research with neural networks was based on synthetic data. This ranges from the earliest experiments with shallow perceptrons and XOR], through Hinton's famous induction of relational structures from family trees (https://www.cs.toronto.edu/~hinton/absps/families.pdf) to very recent criticisms of the learning power of recurrent nets (https://arxiv.org/abs/1711.00350). These studies are useful because the synthetic data exaggerates and emphasises the learning problems that a model would face in the real world, putting the focus on to the important challenges. In this regard, we feel that the simulated environment that we have developed provides a state-of-the-art balance between realism and control for studying grounded language learning in artificial agents (given current technology).
>
> Your concern:
>
> There are a couple of other misunderstandings about the paper that we wanted to clarify. You said:
>
> Although combining LSTM with CNN via a “mixing” module was interesting, it added another layer of complexity that made it more difficult to assess what the results meant....If we need to create a novel deep learning method to illustrate its efficacy, how will it be useful for solving the problem that motivated everything..
>
> Our response:
>
> This is not correct. The paper states:
>
> A mixing module M determines how these signals are combined before they are passed to a LSTM action module A: here M is simply a feedforward linear layer operating on the concatenation of the output from V and L.
>
> We combine visual and language information through concatenation, which is the simplest  (and most general) way of combining this information. Indeed, the guiding motivation behind the model, given our purpose, was that it is as simple as a model can be given our ultimate objective of combining language, motion and vision.
>
> Your concern:
>
> One analysis that would have helped convince me is a comparison to an equivalent non-grounded deep learning model (e.g., a CNN trained to make equivalent classifications), and show how this would not help us understand human behavior.
>
> Our response:
>
> We are a bit worried here that you may have misunderstood the point of this paper. The objective of the work is to understand artificial models - it is not to understand human behaviour. If there was anywhere in the paper that gave the impression otherwise, please let us know and we will correct it immediately!
>
> Thanks again for your time, we value your criticism. We'd really appreciate it if you could give the paper another reading, and reconsider your judgement, having considered our responses.

---

### Official Review · AnonReviewer3 · 2017-11-28
**Interesting work that could be grounded more strongly in cognitive science**

**Rating:** 7
**Confidence:** 4

**Review:**

This paper presents an analysis of the properties of agents who learn grounded language through reinforcement learning in a simple environment that combines verbal instruction with visual information. The analyses are motivated by results from cognitive and developmental psychology, exploring questions such as whether agents develop biases for shape/color, the difficulty of learning negation, the impact of curriculum format, and how representations at different levels of abstraction are acquired. I think this is a nice example of a detailed analysis of the representations acquired by a reinforcement learning agent. The extent to which it provides us with insight into human cognition depends on the degree to which we believe the structure of the agent and the task have a correspondence to the human case, which is ultimately probably quite limited. Nonetheless the paper takes on an ambitious goal of relating questions in machine learning in cognitive science and does a reasonably good job of analyzing the results.

Comments:

1. The results on word learning biases are not particularly surprising given previous work in this area, much of which has used similar neural network models. Linda Smith and Eliana Colunga have published a series of papers that explore these questions in detail:

http://www.iub.edu/~cogdev/labwork/kinds.pdf
http://www.iub.edu/~cogdev/labwork/Ontology2003.pdf

2. In figure 2 and the associated analyses, why were 20 shape terms used rather than 8 to parallel the other cases? It seems like there is a strong basic color bias. This seems like one of the most novel findings in the paper and is worth highlighting.

This figure and the corresponding analysis could be made more systematic by mapping out the degree of shape versus color bias as a function of the number of shape and color terms in a 2D plot. The resulting plot would show the degree of bias towards color.

3. The section on curriculum learning does not mention relevant work on “starting small”  and the “less is more" hypothesis in language development by Jeff Elman and Elissa Newport:

https://pdfs.semanticscholar.org/371b/240bebcaa68921aa87db4cd3a5d4e2a3a36b.pdf
http://www.sciencedirect.com/science/article/pii/0388000188900101

4. The section on learning speeds could include more information on the actual patterns that are found with human learners, for example the color words are typically acquired later. I found these human results hard to reconcile with the results from the models. I also found it hard to understand why colors were hard to learn given the bias towards colors shown earlier in the paper.

5. The section on layerwise attention claims to give a “computational level” explanation, but this is a misleading term to use — it is not a computational level explanation in the sense introduced by David Marr which is the standard use of this term in cognitive science. The explanation of layerwise attention could be clearer.

Minor:

“analagous” -> “analogous”

The paper runs longer than eight pages, and it is not obvious that the extra space is warranted.

---

> ### Author Response · Authors · 2017-12-29
> **We have added better acknowledgement of related cognitive science**
>
> Thank you for these very useful pointers to literature in human language learning. We have added citations to Smith and Colunga as well as to Elman and Newport. Overall, we tried to keep the discussion of human learning to a minimum as the objective was to better understand a class of (artificial) computational model. This is why we do not in general compare the observations made about the network to effects identified in humans. However, our experiments were certainly inspired by this long history of principled experimentation on humans so we want to credit relevant work, while highlighting a relatively new (and potentially vast) application of the same experimental techniques. Please say if we have missed any other studies in human learning that we should mention.
>
> As you suggest, we have replaced the phrase 'computational level' to avoid confusion. Thank you for your review and useful suggestions. If you think the paper is worthy of acceptance, please also liaise with the other reviewers and consider our comments to their criticisms to try to help them see the merits of this research.

---

### Decision · Program_Chairs · 2018-01-29
**ICLR 2018 Conference Acceptance Decision**

**Decision:**

Reject

**Comment:**

This paper resulted in significant discussion -- both between R2 and the authors, and between the AC, PCs, and other solicited experts.

The problem of language grounding (and instruction following) in virtual environments is clearly important, this work was one of the first in the recent resurgence, and the goal of understand what the agents have learned is clearly noble and important. In terms of raw recommendations, the majority reviewer recommendation is negative, but since concerns raised by R2 seemed subjective (which in principle is not a problem), out of abundance of caution, we solicited additional input. Unfortunately, we received feedback consistent with the concerns raised here:

-- The lack of generality of the behavior found. Even if we ignore the difficult question of why the agent prefers what it does, it's unclear how the conclusions here generalize much farther than the model and environment used; the manuscript does not provide any novel or transferable principals of the form "this kind of bias in the environment leads to this kind of bias in models with these properties".

-- We realize even providing that concrete a statement might be hard, but also missing are thorough comparisons to other kinds of models (e.g. non-deep, as asked by R1) to establish that this is a general phenomenon.

Ultimately, there is a sense that this is too narrow an analysis, too soon. If there was one architecture for learning embodied agents in 3d environments that was clearly successful and useful, then studying its properties might be interesting (even crucial).  But the dust in this space isn't settled. Our current agents are fairly poor, and so the impact of understanding the biases of a specific model trained in a specific environment seems fairly low.

Finally -- this not taken into consideration in making the decision -- it is not okay to list personal homepage domains (that may reveal author identity to ACs) as conflict domains; those are meant for institutional conflicts/domains.